# Development, Chlorophyll Content, and Nutrient Accumulation in In Vitro Shoots of *Melaleuca alternifolia* under Light Wavelengths and 6-BAP

**DOI:** 10.3390/plants13202842

**Published:** 2024-10-11

**Authors:** Antony Cristhian Gonzales-Alvarado, Jean Carlos Cardoso

**Affiliations:** 1Master Science Graduate Program of Plant Production and Associated Bioprocesses, Center of Agricultural Sciences, Federal University of São Carlos, Araras 13600-970, SP, Brazil; 2Laboratory of Plant Physiology and Tissue Culture, Department of Biotechnology, Plant and Animal Production, Center of Agricultural Sciences, Federal University of São Carlos, Araras 13600-970, SP, Brazil

**Keywords:** tea tree, light-emitting diodes, cytokinin, development, shoot multiplication, nutrition

## Abstract

In vitro cultivation of *Melaleuca* could contribute to the cloning of superior genotypes. Studies of factors affecting micropropagation are needed, such as the interaction with light-emitting diodes (LEDs) and plant growth regulators added to the culture media. This study aimed at better understanding the effects of spectra on the development and physiology of melaleuca cultivated in vitro, as well as the interaction of LEDs with the main cytokinin used in micropropagation, N6-Benzylaminopurine (6-BAP). 6-BAP, spectra, and their interaction had a significant effect on most of the variables analyzed, altering the in vitro development and chlorophyll concentrations in the plants, as well as changing different variables in the culture medium, such as pH, EC, and levels of Ca^2+^, Mg^2+^, and P, and nutrient accumulation in the shoots. The results demonstrate that the main effects of adding BAP to the in vitro cultivation of melaleuca are an increase in the number of shoots, which resulted in greater fresh and dry masses; a reduction in height and chlorophyll content; complete inhibition of adventitious rooting; higher consumption of Mg, and lower consumption of Ca and P from the culture medium; higher content of Fe, and lower content of P, S, Mn, Cu and B in the in vitro shoot tissues.

## 1. Introduction

The genus *Melaleuca*, belonging to the Myrtaceae family, includes around 150 species [1] and is used to produce essential oils on an industrial scale, particularly the species *Melaleuca alternifolia* Cheel. This species has different properties, such as anti-inflammatory [2,3], antimicrobial [3,4], acts as a fungicide [3,5] and acaricide [6], and has a known effect for the treatment of insect bites and skin infections [3,7].

There is great interest in developing micropropagation protocols to evaluate the feasibility of this technique, from the commercial point of view, and the performance of plantlets from the micropropagation system, since melaleuca seedlings results in high heterogeneity and stem cutting has difficulties such as low rooting percentage [8,9,10]. Despite the limited number of studies, they have revealed the substantial potential of utilizing micropropagation as a clonal system for the production of melaleuca plantlets [11,12]. In these studies, the use of the cytokinin N6-Benzylaminopurine (6-BAP) has proven to be essential in achieving high rates of multiplication and proliferation of shoots under in vitro conditions [11,12,13] by promoting the proliferation of axillary and adventitious buds in tissues cultured in vitro [14].

Advances in indoor cultivation including in vitro conditions are favored by light-emitting diodes (LEDs), which replaced cold or fluorescent lamps, increasing energy efficiency, and expanding the types and combinations of LED spectra according to each growing system or plant species cultivated [15]. Also, using LED systems emerges as an extremely promising strategy to modulate the physiological response of plants [16]. This approach allows the generation of a specific and combined wavelengths, which has been shown to induce particular responses in plants, notably an increase in photosynthetic rate, and significantly modulate plant growth [17].

Currently, the effects of LEDs with different wavelengths (spectra) are largely explored in the in vitro development of diverse plant species [18]. In *Vitis riparia*, a significant increase in shoot height was achieved when using red light, which also increased shoot biomass [19]. In *Helianthus tuberosus*, blue light favored an increase in biomass and the growth of explants overall [20], while in *Gossypium hirsutum*, *Cedrela fissilis*, and *Pinus pseudostrobus*, the combination of red and blue lights increased shoot production, elongation, and biomass accumulation [21,22,23]. Given these different results, the establishment of optimal conditions for in vitro cultivation using LEDs is required to improve the growth. Most studies on in vitro micropropagation of *Melaleuca* have used conventional cold lamps or LEDs with specific wavelengths of white and red/blue light combined (1:1), respectively [11,12,24].

Therefore, there is a lack of studies demonstrating the effects of different wavelengths provided by LEDs on the in vitro development of melaleuca. In addition, there is no knowledge of the interaction of the different light wavelengths combined with phytoregulators, such as 6-BAP. This cytokinin is widely used for micropropagation of several commercial and non-commercial species under in vitro conditions. Finally, few studies have been dedicated to better understanding the chemical changes in culture media in response to the development and cultivation of plants in vitro. This information can help to improve comprehension of the performance of plants under in vitro conditions or to determine key factors that enhance the performance of the culture media [25].

Thus, the main objectives of this study were to better understand the interaction of the cytokinin 6-BAP and the different wavelengths provided by LEDs on the in vitro development of melaleuca shoots, as well as the main changes in the culture media associated with the nutritional features. At the same time, this study also aimed at identifying the best wavelengths associated with the proliferation/multiplication of melaleuca under in vitro conditions.

## 2. Results

### 2.1. Effects of 6-BAP and Light Spectra on the Cultivation and Development of Melaleuca In Vitro

The effect of BAP, LED spectra, and their interaction was significant (*p* ≤ 0.05) for the following variables analyzed: number and length of new shoots, shoot height, chlorophyll a, chlorophyll b, and total chlorophyll (CLT = Chla + Chlb) content in leaves. Regarding the fresh and dry masses, the effect of BAP or LED spectra was significant, but no interaction between these factors was observed (Table 1). Our results demonstrated the strong effect of 6-BAP on shifting the type of development of melaleuca under in vitro conditions (Figure 1).

The highest proliferation of shoots (8.1 shoots/nodal segment) was obtained using white LEDs in the presence of 6-BAP, while in its absence, there were only 2.5 shoots/nodal segment. The opposite was found for new shoot length and height, in which the addition of 6-BAP resulted in a drastic reduction, from 1.17 cm (−BAP) to 0.54 cm (+BAP), when cultivated in blue + red light (BR). Similarly, using white LEDs, shoot height was drastically reduced from 7.3 cm to 1.8 cm, in the presence of 6-BAP. Interestingly, the highest fresh and dry masses of explants (180 mg/shoot and 15 mg/shoot, respectively) were obtained using white light in the presence of 6-BAP, with low mass values in the absence of 6-BAP. Except for blue light and cultivation in dark conditions, white light did not differ statistically from red, red + white, and red + blue for most characteristics associated with in vitro melaleuca development.

The chlorophylls a and a + b contents were higher (Chla, 39.17 mg L^−^^1^ to 49.27 mg L^−^^1^; Chla+b, 73.08 mg L^−^^1^), in the absence of 6-BAP. With the addition of 6-BAP, the values were 16.12 mg L^−^^1^ to 26.17 mg L^−^^1^ (Chla) and 28.26 mg L^−^^1^ to 44.27 mg L^−^^1^ (Chla+b), respectively (Table 1). The light spectra (except for dark × light conditions) had a minor influence on chlorophyll contents compared to 6-BAP effects, but white light resulted in higher contents of Chla and Chla+b compared to blue + red light.

### 2.2. Effect of 6-BAP and Light Spectra on the Density and Diameter of Leaf Stomata (DEE and DIE)

The effect of 6-BAP on the density and diameter of stomata was dependent on the light spectra used for melaleuca cultivation (Table 2). The highest density of stomata (235/mm^2^) was obtained when melaleuca shoots were cultivated under red + white in the presence of 6-BAP, while with the same light source and absence of 6-BAP, the observed density of stomata was only 173.3/mm^2^. Under white and red + white lights, the density of stomata was higher when 6-BAP was added to the culture media, while under monochromatic red light, the density of stomata was higher in the absence of 6-BAP. Except for white light, where the density of stomata was higher in the presence of 6-BAP, in most of the other lights, the absence of 6-BAP resulted in larger (B, R, W + R) or equal (B + R) density of stomata (Table 2).

### 2.3. Effect of 6-BAP and Wavelength on Some Chemical Characteristics of the Culture Media

All chemical characteristics of the culture media were affected by the presence or absence of 6-BAP as well as the light wavelength used for in vitro cultivation. The presence of BAP seems to enhance the chemical dynamics between the culture media and the cultivated shoots. In five out of six light wavelengths tested, the pH reduction was more pronounced when 6-BAP was present, with a decrease of −0.8 to −1.2 points compared to the initial pH. The smallest variations in pH during in vitro cultivation were observed under dark conditions and in the red or white LEDs, in the absence of 6-BAP, with a −0.3 to −0.4 point reduction compared to the initial pH (Table 3).

Different from pH, the most intense changes in electrical conductivity (EC) were attributed to light wavelengths. As an example, the highest variations in EC were observed in culture media incubated under white light in the absence of 6-BAP (1.13 mS/cm) and blue light (0.73–0.91 mS/cm), respectively. The greatest variation in EC occurred in the absence of 6-BAP under white LEDs and in the presence of 6-BAP under blue LEDs (Table 3).

Regarding the uptake of nutrients from the media, a greater consumption of calcium after in vitro development of melaleuca was observed in the cultivation under monochromatic red (63.9%) and red + white LEDs (68.9%) in the absence of 6-BAP. This cytokinin had a strong influence on calcium, generally reducing its uptake when shoots were cultivated under white, red, and red + white LEDs (Table 3). Similar results were reported for P, for which there was a higher consumption of P (42.9 to 51.0%) in the absence of BAP in the culture medium. The highest percentage of Mg consumption (20.6–23.2%) occurred in red light, regardless of the presence of 6-BAP, and in the red + white LEDs in the presence of 6-BAP.

The correlation analysis detected significant relationships between the variables studied. The number of shoots was strongly associated with biomass, in both fresh and dry masses, and presented an inverse relationship with shoot length. In turn, shoot length showed a strong correlation with chlorophyll content. In addition, a correlation was identified between stomata density and culture medium conditions, such as pH and electrical conductivity, suggesting an influence of the growth environment on stomatal regulation and nutrient uptake (Figure 2).

Principal component analysis (PCA) revealed some interesting and new information about how factors such as 6-BAP and light wavelength affected the variables analyzed (Figure 3). Among these, increases in stomatal density, the number of shoots, and fresh and dry masses were observed in the presence of 6-BAP, while increases in chlorophyll contents, shoot length, and absorption of Ca and P were observed in the absence of 6-BAP in the culture medium.

### 2.4. Effects of 6-BAP and Light Spectra on Nutrient Content in Shoot Tissues

The presence of 6-BAP in the culture medium had a significant effect on seven out of the ten nutrients analyzed in melaleuca tissues cultivated in vitro (Table 4). The concentrations of P, S, Mn, Cu, and B were mostly lower in tissues cultivated in the presence of 6-BAP added to the culture medium, while the Fe concentration was higher in the presence of 6-BAP. Regarding the wavelengths tested, the dark condition was the only treatment in which the presence of 6-BAP promoted a greater absorption of all analyzed nutrients compared to the medium in the absence of 6-BAP. In the light spectra with better growth of melaleuca (white and white + red), except for Fe, all other nutrients had lower contents in the tissues when 6-BAP was added to the culture media.

## 3. Discussion

### 3.1. Effects of 6-BAP and Light Spectra on In Vitro Development of Melaleuca

In the present study, a multiplication factor of 5.6 to 8.1 shoots/nodal segment was reported with the addition of a low concentration of 6-BAP (0.25 mg L^−^^1^) to the culture medium. According to Iiyama and Cardoso [11], the addition of 6-BAP at a low concentration (0.55 μM = 0.124 mg L^−^^1^) causes a considerable increase in shoot proliferation, up to 6.54 shoots/explant. Similar results were reported by Oliveira et al. [12], with the highest proliferation of shoots (5.6 shoots/nodal segment) obtained with the addition of 0.55 μM of 6-BAP to the MS culture medium. In fact, the main physiological effect of 6-BAP on the in vitro cultivation of melaleuca was the induction of multiple shoots in vitro [11], by increasing the number of lateral buds and the proliferation of new shoots [26]. However, 6-BAP had harmful effects on in vitro melaleuca development, strongly decreasing shoot length and height and inhibiting root development in shoots (Figure 1). These negative effects were also reported by Iiyama and Cardoso [11], who observed that 6-BAP-derived micro-shoots developed normally when subcultured in the rooting/elongation medium in the absence of 6-BAP. Fresh and dry masses showed a strong correlation with increases in the number of shoots promoted by 6-BAP addition to the media (8.1 shoots/explant) (Figure 2). In different species, this same correlation was found between fresh and dry masses with the increase in the number of shoots/explants [23,27,28,29]. Another interesting effect of 6-BAP in the culture media is the maintenance of growth, even reduced, under dark conditions. In the absence of 6-BAP, the plantlets died before completing 30 days of cultivation. 

The concentrations of chlorophylls a and total (a + b) were higher in the shoots obtained from plantlets cultivated in the absence of 6-BAP, regardless of the light wavelength. 

Regarding the effects of light spectra, there were no differences in chlorophyll contents in the absence of 6-BAP. According to Lotfi et al. [30], blue light is relevant for chlorophyll biosynthesis, chloroplast development, and stomatal opening [31], and similar results were obtained by other authors in different plant species [22,32]. However, there are contrasting results with other cultures such as bananas in which the use of white + red and red LEDs resulted in higher concentrations of chlorophylls, compared to the use of fluorescent light [33]. 

Conversely, using the same light source, the presence of 6-BAP in the culture medium resulted in a significant reduction in the concentrations of chlorophyll a and total chlorophyll but did not influence chlorophyll b content. These results are similar to those observed for other species. In *Nidularium* (Bromeliaceae) in vitro cultivation, an increasing concentration of 6-BAP and other cytokinins in the culture medium resulted in a significant reduction in chlorophyll concentrations in plant tissues [34]. An increase in chlorophyll levels in shoots was reported in transgenic *Arabidopsis thaliana* under reduced cytokinin levels [35].

The highest stomatal density was obtained with the combination of red + white LEDs, followed by red, blue + red, and white LEDs, while in monochromatic blue light, there was a significant decrease in stomatal density. However, He et al. [36] observed that the highest average density of stomata (10.9 stomata per 40,000 µm^2^) was achieved by growing *Camellia oleifera* under white LED light, and Mihovilović et al. [37] reported the highest stomatal density (17.3 stomata per mm^2^) of *Amelanchier alnifolia* when cultivated under blue + red LEDs. Interestingly, melaleuca cultivated in vitro under blue LED light presented the lowest density of stomata, regardless of the presence of 6-BAP. This is the opposite of that observed in cucumber seedlings, in which a relative reduction of red light and an increase of blue light resulted in the highest stomatal density [38]. Izzo et al. [39] reported a reduction of stomatal density both under red or blue monochromatic light, while the mixed light increased stomatal density.

Under in vitro conditions, the stomata are normally round and open [40], and these characteristics were also observed in the present study with melaleuca shoots. The largest diameters of leaf stomata (26.5 μm and 24.8 μm) were observed under blue or red monochromatic LED lighting, both in the presence of BAP and under white light in the absence of BAP. Similar results were obtained in three species (*Cordyline australis*, *Ficus benjamina*, and *Sinningia speciosa*) by Zheng and Van [41], who reported that the use of blue or red LEDs resulted in the largest diameters of stomata. Cioć and Pawlowska [42] reported that the addition of 6-BAP to the culture medium resulted in longer stomata and, consequently, larger diameters. This study with melaleuca, however, demonstrated that this increase in stomatal diameter in response to the addition of 6-BAP to the culture medium depended on the wavelength, being higher with the blue, red, and red + white LEDs. These differential responses to light according to the species are the result of several factors affecting stomata development such as the species group, light intensity, the stage of development, and photosynthetic metabolism, among others [43], and others not related to light such as the osmotic potential of the culture media [11].

One of the main problems with melaleuca micropropagation is the acclimatization stage, in which there is rapid plantlet dehydration, which is partially attributed to stomatal density, formation, and functionality [11]. Thus, the LED lighting influence on stomata could be used to diminish the stomatal density and diameter for better preparation of plantlets for acclimatization. Changes in the in vitro environmental conditions are strongly associated with improvements in the acclimatization stage, and stomata characteristics and functionality are major factors affecting the success of the acclimatization stage, especially in tree species [44].

### 3.2. Effects of 6-BAP and Light Spectra on Changing the Culture Media Characteristics 

Although in vitro studies have explored extensively the influences of culture media, additives, and plant growth regulators on plant development and propagation, little attention has been given to changes in these culture media in response to the development of in vitro tissues. Changes in culture media in response to in vitro cultivation systems are influenced by several factors such as the system used for in vitro cultivation. Kozai et al. [45] reported that photoautotrophic conditions (no sugar in culture media and a CO_2_-enriched environment) promoted the highest uptake of PO_4_^3−^, NO_3_^−^, Ca^2+^, Mg^2+^, and K^+^. In the present study, the highest ion uptakes of nutrients P and Ca also seem to have occurred in the absence of 6-BAP in the culture medium, especially when grown in red and red + white LEDs. The opposite was observed for Mg^2+^ uptake, in which the presence of BAP resulted in higher uptake of this nutrient under most spectra used. Gerovac et al. [46] and Schroeter-Zakrzewska and Kleiber [47] reported the highest calcium and magnesium uptake under red LED light. 

The largest range in pH of the culture media after growing melaleuca in vitro occurred in the presence of 6-BAP, especially when melaleuca shoots were cultivated under blue or red monochromatic LEDs (−1.1 to −1.2) and when shoots were cultivated under blue + red LEDs. For electrical conductivity, the response was contrary to pH, and the greatest variations in EC occurred in the absence of 6-BAP added to the culture medium. The smallest variations in pH and EC, compared to the initial medium, were observed in dark conditions and without 6-BAP, demonstrating that the absence of light for in vitro melaleuca resulted in limited development of the shoots/plantlets and few variations in the culture media. Perez-Vazquez et al. [48] reported that the EC in the culture media is correlated with the development of morrón pepper, presenting a correlation between a decrease in EC and a increase in shoot growth, by increasing fresh and dry masses. Nevertheless, the results with melaleuca did not demonstrate this correlation between a drop in EC and a greater accumulation of fresh and dry masses, indicating that other factors related to the type of plant development, such as the formation of plantlets (shoots and roots: absence of 6-BAP) or only shoots (presence of 6-BAP) and the higher growth in the height of plantlets in vitro (absence of 6-BAP), were associated with increases in the uptake of specific nutrients, such as Ca and P, and probably other components of the culture medium that mainly contribute to EC values.

Correlation analysis showed that a limited number of variables analyzed in melaleuca demonstrated significant correlations at 5 (*) or 1% (**) (Figure 2). Among these, the increase in the number of shoots, promoted by the presence of BAP, was highly correlated with the increase in fresh (r = 0.81 **) and dry (r = 0.85 **) masses of shoots and the reduction in shoot size. Also, the higher shoot height, promoted by the absence of BAP, was correlated with increased chlorophyll a (r = 0.77 *) and total chlorophyll (r = 0.77 *) concentrations in the leaves. The presence of 6-BAP is often correlated with a decrease in photosynthetic pigments in different species cultivated in vitro [49]. For culture media, there was a correlation between the higher values of final pH (small variations in relation to initial pH), the higher uptake of P by the shoots (r = 0.71 *), and the height of shoots (r = 0.71 *), suggesting that higher uptake of P increased the shoot height of melaleuca. However, is not possible to know if this effect is direct or indirect, as 6-BAP only promotes shoot induction, while in its absence, shoots and roots were observed in melaleuca in vitro (Figure 1).

Principal component analysis (PCA) revealed that the LED-wavelength component (PC1) explained much of the variability of the data representing a proportion of 50.29%, while the component 6-BAP (PC2) contributed 14.56% to the explanation of data variation (Figure 3). PCA analysis showed that the number of shoots and the fresh and dry masses were highly correlated with the presence of 6-BAP in the culture medium, especially under white or red LEDs.

### 3.3. Effects of 6-BAP and Light Spectra on the Concentration of Nutrients in Tissues

The presence of cytokinin 6-BAP increased substantially the contents of all nutrients in in vitro tissues of melaleuca under dark conditions (+62.9% to +341.6% compared to BAP-free media). Cytokinins like 6-BAP are well known to delay tissue senescence under dark conditions, by maintaining higher transcript levels of Photosystem-II-related genes and the integrity of chlorophyll pigment-protein, maintaining Chl a/b ratios [50]. The addition of 6-BAP and other cytokinins also induces photomorphogenesis under dark conditions [51]. Thus, cytokinins applied exogenously can maintain active metabolism and morphogenesis, including nutrient uptake of in vitro plantlets, under dark conditions. Indeed, cytokinins can provide nutrient accumulation in tissues under adverse conditions [52].

However, under light conditions, the addition of 6-BAP resulted in decreased accumulation of nutrients in the tissues for most nutrients analyzed, especially phosphorus (P), magnesium (Mg), sulfur (S), manganese (Mn), copper (Cu), and boron (B), compared to the tissues cultivated in 6-BAP-free culture media. This is a surprising result since a high amount of fresh and dry mass was observed in the presence of 6-BAP. Interestingly, other authors report that the increased activity of the cytokinin dehydrogenase enzyme, which leads to cytokinin degradation, resulted in an increase of some nutrients in leaves due to the enhancement of the root system and better scavenging of macro- and micronutrients, since cytokinin negatively regulates root growth [35,52]. In fact, in our results with melaleuca, roots in the culture media were only observed in the absence of 6-BAP added to the culture media, which explains the higher accumulation of these nutrients in the tissues cultivated without this cytokinin. 

On the other hand, 6-BAP resulted in significantly higher iron (Fe) concentrations in all samples, regardless of the wavelength of light used. This observation contrasts with that of Séguéla et al. [53], who reported that cytokinins negatively regulate iron uptake in *Arabidopsis* roots; this is because this cytokinin stimulates cell division and growth, which may facilitate higher Fe uptake from the culture medium [54]. However, Nehnevajova et al. [35] reported that only iron, among all other nutrients analyzed, was reduced (11–18%) in the leaves of transgenic lines of oilseed rape plants overexpressing the CKX2 gene, with lower cytokinin content, compared to the wild type. However, most of these studies were associated with the reduction of Fe uptake by roots. In the present study, the use of 6-BAP inhibited root formation in melaleuca explants, and no studies were found exploring the in vitro uptake of Fe and its relationship with cytokinin content. Thus, these studies and our findings with melaleuca suggest a more complex mechanism of regulation of iron uptake in plants, in which cytokinins can play an important role in iron uptake and accumulation in tissues.

## 4. Materials and Methods

### 4.1. Plant Material and In Vitro Growing Conditions

The plant material used for the experiments consisted of in vitro stem nodal segments (shoots) of 1.0–1.5 cm in length, previously established in vitro from the stem apices taken from greenhouse mother plants. These explants were established and propagated in vitro for approximately nine months to obtain sufficient stem nodal segments for the experimental design. These nodal segments were grown in 15 mL of modified Murashige and Skoog (MS) medium [55], containing half the macronutrient concentration (MS ½), 6.5 g L^−1^ agar (Agargel, João Pessoa-PB, Brazil), 2% sucrose, 0.1 g L^−1^ inositol, and with the pH adjusted to 5.8 [11]; with or without the presence of 6-BAP (at 0.25 mg L^−1^) in the culture media. The 6-BAP used was added before the pH adjustment and was obtained from a concentrated solution containing 10 mg 6-BAP (Sigma-Aldrich, St. Louis, MO, USA), previously dissolved in 1 mL NaOH (1 M) followed by addition of 9 mL distilled water.

The test tubes containing individual microcuttings, each one with four nodes and 0.7 ± 0.1 cm (replicates), were kept in a growth room with the photoperiod adjusted to 14 h, at a temperature of 25 ± 2 °C, and illuminated with six different wavelengths provided by LEDs. The flow density of photosynthetic photons (PPFD) was measured for each wavelength using the Quantum SQ-520 m (Apogee Instruments, Logan, UT, USA). The LEDs used as treatments were as follows: (1) 100% white diffuser module (Ourolux^®^, São Paulo, Brazil), with peaks at 440–450 nm (blue), 540–550 nm (green), and 610–620 nm (red), W: 64.09 μmol/m^−2^ s^−1^), (2) dark conditions (D: 0.58 μmol/m^2^s—diffuse light), (3) 100% blue LED light (Phillips Greenpower LED Research Module Blue, ~440 nm, Pila, Poland, B: 55.58 μmol/m^2^s), (4) 100% red LED light (Phillips Greenpower LED module HF deep red, ~660 nm, Pila, Poland, R: 106.22 μmol/m^2^s), (5) 50%: 50% red and white LED light (RW: 104.51 μmol/m^2^s), and (6) 40%: 60% blue and red LED light (LabPar, with wavelength peaks at 447 nm, range 420–470 nm—blue and 667 nm, range 625–680 nm—red, Barueri, Brazil, BR: 71.48 μmol/m^2^s).

### 4.2. In Vitro Characteristics of Melaleuca Cultivated under LED Spectra and 6-BAP

The analyses of melaleuca development were carried out weekly for 30 days of in vitro cultivation. The variables evaluated were the number and length of shoots (cm), shoot height (cm), total fresh and dry masses (g), and shoot mass (g). For the fresh and dry masses analysis, plantlets were divided into shoots and roots (when present) or only shoots (no roots) and weighed on a digital analytical scale accurate to 0.1 mg (ML201, Mettler, Switzerland). The dry mass of each part was obtained by oven-drying at a temperature of 70 °C for 72 h, followed by measurement on the same scale. Chlorophyll a, b, and total chlorophyll contents were analyzed using spectrophotometry. Chlorophyll a, b, and total chlorophyll concentrations were obtained using 100 mg of the in vitro melaleuca leaves. These were macerated in a crucible containing 2–3 mL pure acetone (99.5%) and later adjusted to a final volume of 10 mL for spectrophotometric analysis. The samples were placed in the manufacturer’s glass bottle to measure their absorbance in an IRIS-HI801 spectrophotometer (Hanna Instruments, Woonsocket, RI, USA). Absorbance readings were performed at 664 nm and 647 nm, according to the equations by Lichtenthaler [56] and against a blank of pure acetone. Chlorophyll concentrations were determined in mg L^−1^ using the following equations: chlorophyll a = 12.70 (A 664) − 2.97 (A647); chlorophyll b = 20.70 (A 647) − 4.62 (A 664), as proposed by Lichtenthaler et al. [57]. Total chlorophyll was obtained by summing chlorophylls a and b.

Stomatal size and density were also evaluated in in vitro leaves of melaleuca. For counting and measuring the polar and equatorial diameters of the stomata, leaves from the fourth to the sixth node of the in vitro plantlets were used [11]. The leaves were first fixed in Carnoy solution composed of three volumes of ethanol and one of glacial acetic acid (*v*/*v*) for 48 h. Subsequently, samples were stored in 70% alcohol at 8 °C until the moment of preparation of the slides. For the visualization of the stomata, the leaves were previously immersed in 70% alcohol for one minute, and soon after in potassium hydroxide solution 5 M at 45 °C for 20–30 s, and the digestion reaction was stopped with distilled water. Slides were mounted with coverslips and analyzed with a light microscope Nikon Eclipse 201 (Nikon, Melville, NY, USA), with a total magnification of 100×, coupled to a high-resolution Opticam camera (5 Mp) (Opticam, Sao Paulo, Brazil) with software for the measurement of the stomata. The variables evaluated were stomatal density and diameter (µm) in 10 observed fields per leaf.

### 4.3. Chemical Characteristics of Culture Media Used for Melaleuca In Vitro Cultivation

The purpose was to observe the differences between the initial and final values of different characteristics of culture media such as pH, EC, and some nutrients, to investigate the effects of the treatments (LEDs and 6-BAP) on the chemical changes of the culture medium after melaleuca development. The initial (after autoclaving and before cultivation) and final (after in vitro cultivation for 30 days) values of pH and EC of the culture medium were measured using a pH meter (SevenCompact, Mettler, Scwerzenbach, Switzerland) and a portable conductivity meter (Hanna Instruments, Amorim, Portugal). The final and initial pH and EC data were used to obtain their variation throughout the in vitro cultivation period. For these analyses, the culture media were previously frozen at −20 °C for 24 h, followed by thawing, and separation of liquid phases for determinations using the pH meter and conductivity meter.

The analysis of macronutrient consumption was as follows: Ca^2+^—using 3 mL of sample, 10 mL of reagent HI93752A-Ca, 4 drops of buffer, and 1 mL of HI93752B-Ca, reading the absorbance at 466 nm; phosphate (high range)—10 mL of sample, 10 drops of HI93717A-0 and one packet of HI93717B-0, reading the absorbance at 525 nm; and Mg^2+^—using 1 mL of HI93752A-Mg, 10 mL of HI93752B-Mg, and 0.5 mL of sample, reading the absorbance at 466 nm. The sample used was from the culture media; the initial and final values of the nutrients were also analyzed, aiming to determine the amounts of nutrients absorbed from the culture media. The concentrations (mg L^−1^) of these nutrients from the culture media were obtained using a IRIS-HI801 spectrophotometer (Hanna Instruments, Woonsocket, USA), using the methodologies and solutions provided by the manufacturer (HI801 instruction manual, Hanna Instruments, 2020, Woonsocket, USA)

### 4.4. Nutrient Analysis in Melaleuca Shoot Tissues 

Shoot tissue samples were dried in an oven at 60 °C for 48 h. Afterward, they were ground into a fine powder and sent to the soil and plant analysis laboratory located at Instituto Agronômico de Campinas (IAC), Campinas, Brazil. The analyzed nutrients were phosphorus (P), potassium (K), calcium (Ca), magnesium (Mg), sulfur (S), iron (Fe), manganese (Mn), copper (Cu), zinc (Zn), and boron (B), using nitric-perchloric digestion and analyzed by inductively coupled plasma-optical emission spectroscopy (ICP-OES) [58].

### 4.5. Experimental Design and Data Analysis

The experiment was conducted in a factorial arrangement: two concentrations of 6-BAP (0.0 mg L^−1^ and 0.25 mg L^−1^) and six light spectra (W, D, B, R, WR, and BR), with ten repetitions per treatment, and the experimental unit (repetition) was a test tube containing one nodal segment (1.0–1.5 cm). The experiment was repeated twice.

All the results obtained were analyzed by two-way analysis of variance (ANOVA) and when they presented normality (Shapiro–Wilk), the means were compared by Duncan’s new multiple range test at *p* ≤ 0.05 significance. When the data were not normally distributed and could not be normalized by data transformation, they were tested using the Games–Howell test, also at *p* ≤ 0.05 significance. For these analyses, R Studio software, version 4.3.2 [59] was used. At the end of the evaluations, correlation analysis and principal component analysis were run for all the variables analyzed.

Additionally, the data was evaluated using correlation and principal component analysis, where one of the factors was modified, where one treatment (dark condition) was removed to evaluate the fifteen variables of the melaleuca development, as melaleuca does not grow in the dark, thus using a 2 × 5 factorial. The correlation was run with Spearman at the level of 5% probability.

## 5. Conclusions

There is a strong interaction between the addition of cytokinin 6-BAP to the culture medium and the wavelengths provided by LEDs used for the in vitro cultivation of melaleuca. The use of 6-BAP had important effects; it increased the number of shoots/explants, resulting in increased fresh and dry masses, reduced shoot height, and significantly decreased the chlorophyll contents of the plants in vitro. White or white + red LEDs are recommended for growing melaleuca, aiming at its multiplication in vitro with 6-BAP-supplemented media or its use in rooting and elongation of shoots without 6-BAP supplementation. 6-BAP also negatively affected the nutrient concentrations in the plant tissues, except Fe, which increased substantially independent of light wavelength.

## Figures and Tables

**Figure 1 plants-13-02842-f001:**
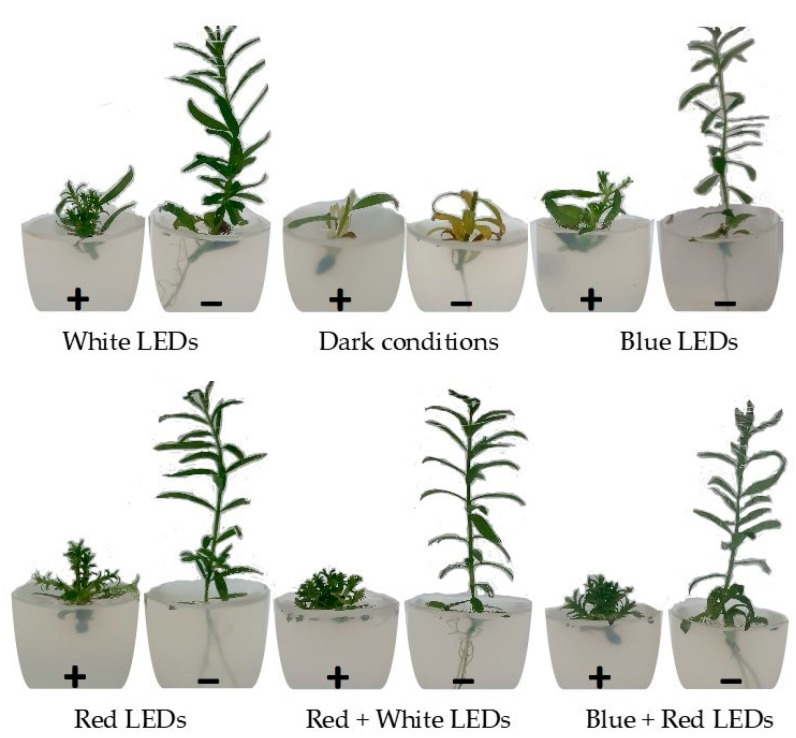
In vitro development of melaleuca in culture media containing 6-BAP (+) or not (−) and grown under different light spectra.

**Figure 2 plants-13-02842-f002:**
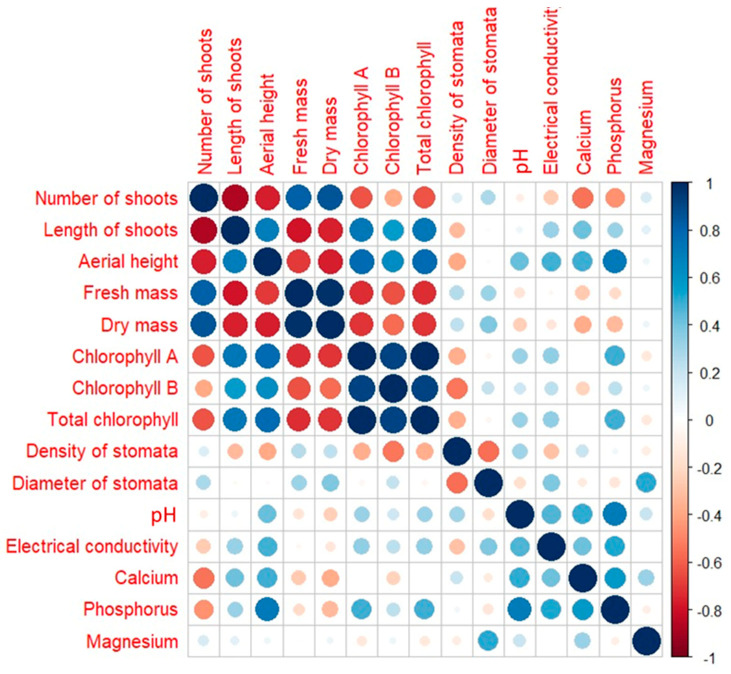
Correlation analysis between several characteristics of melaleuca shoots and plantlets cultivated under different LED wavelengths and the presence of 6-BAP. The increasing in diameter of the circles represents the increase in correlation of the analyzed variables.

**Figure 3 plants-13-02842-f003:**
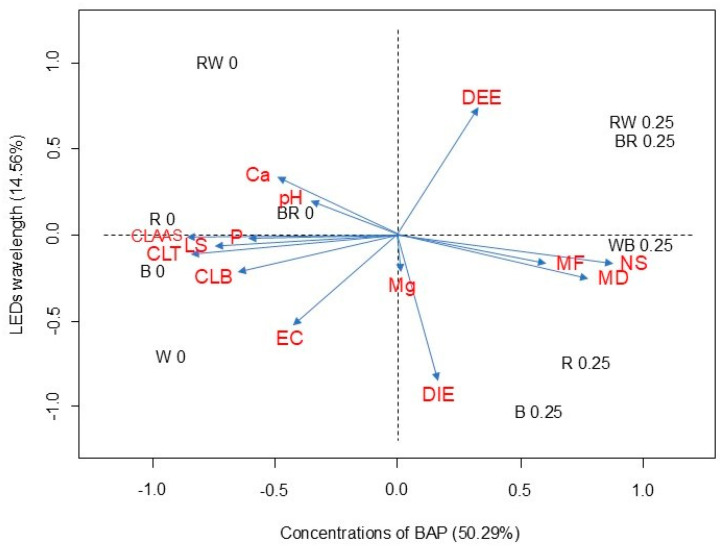
Principal component analysis (PCA) for 6-BAP concentrations and LED wavelengths in the in vitro cultivation of melaleuca. Mg, P, Ca, EC, pH, DIE, DEE, CLT, CLB, CLA, MD, MF, AS, LS, and NS (magnesium, phosphorus, calcium, electrical conductivity, pH, diameter of stomata, density of stomata, total chlorophyll, chlorophyll b, chlorophyll a, dry mass, fresh mass, shoot height, shoot length, number of shoots).

**Table 1 plants-13-02842-t001:** Effect of 6-BAP and light spectra on in vitro development of melaleuca.

6-BAP (mg L^−1^)	Light Spectra					
	White	Dark Conditions	Blue	Red	Red + White	Blue + Red
Number of shoots						
0	2.5 Ba	0.01 Bc	1.88 Bab	2.0 Bab	1.5 Bb	1.7 Bab
0.25	8.1 Aa	5.8 Aa	5.6 Aa	6.9 Aa	6.2 Aa	7 Aa
New shoot length (cm)						
0	0.93 Aa	0.02 Bb	1.08 Aa	1.15 Aa	0.9 Aa	1.17 Aa
0.25	0.48 Ba	0.79 Aa	0.85 Aa	0.63 Ba	0.64 Aa	0.54 Ba
Shoot height (cm)						
0	7.3 Aa	2.2 Ab	5.6 Aab	5.4 Aab	5.7 Aab	4.9 Aab
0.25	1.8 Ba	2.2 Aa	2.5 Ba	2.2 Aa	1.7 Ba	1.8 Ba
Fresh mass (mg/shoot)						
0	80 Ba	30 Bb	50 Ba	60 Aa	60 Ba	60 Ba
0.25	180 Aa	90 Ab	110 Aab	80 Ab	100 Ab	90 Ab
Dry mass (mg/shoot)						
0	10 Ba	4 Bb	7 Bab	9 Ba	8 Ba	9 Ba
0.25	15 Aa	7 Ab	13 Aa	12 Aa	13 Aa	12 Aa
Chlorophyll a (mg L^−1^)						
0	49.17 Aa	-	49.27 Aa	48.33 Aa	39.17 Aa	40.17 Aa
0.25	26.17 Ba	-	16.12 Aab	21.3 Bab	17.62 Bab	12.74 Bb
Chlorophyll b (mg L^−1^)						
0	20.7 Aa	-	23.8 Aa	20.61 Aa	18.04 Aa	18.8 Aa
0.25	18.1 Aa	-	12.14 Ba	19.04 Aa	12.47 Aa	10.17 Aa
Total chlorophyll (a + b) (mg L^−1^)						
0	69.87 Aa	-	73.08 Aa	68.95 Aa	57.21 Aa	58.97 Aa
0.25	44.27 Ba	-	28.26 Bab	40.34 Bab	30.09 Bab	22.91 Bb

Different capital letters (columns) indicate a significant difference between absence and presence of 6-BAP and different lowercase letters (rows) represent the differences between the light spectra, according to the Duncan test; *p* < 0.05.

**Table 2 plants-13-02842-t002:** Effect of 6-BAP and light spectra on the density and diameter of stomata.

6-BAP (mg L^−1^)	Light Spectra				
	White	Blue	Red	Red + White	Blue + Red
Stomatal density (number of stomata/mm^2^)					
0	116.6 Bb	80 Ac	140 Ab	173.3 Ba	135 Ab
0.25	155 Ab	81.6 Ac	115 Bc	235 Aa	163.3 Ab
Stomatal diameter (μm)					
0	22.5 Aa	18.4 Bc	20.2 Bb	15.1 Bd	18.3 Ac
0.25	18.4 Bb	26.5 Aa	24.8 Aa	18.7 Ab	17 Ab

Different capital letters (columns) demonstrate a significant difference between absence and presence of 6-BAP and different lowercase letters (rows) represent the differences between the light spectra, according to the Duncan test; *p* < 0.05.

**Table 3 plants-13-02842-t003:** Effects of 6-BAP and wavelength on the final culture media pH, EC, and the macronutrient (Ca, Mg, and P) uptake from the culture media.

6-BAP (mg L^−1^)	Light Spectra					
	White	Dark Condition	Blue	Red	Red + White	Blue + Red
pH final (pH final–initial)						
0	4.9(−0.4) AaB	5.0(−0.3) Aa	4.5(−0.8) Abc	5.0(−0.4) Aab	4.7(−0.7) Aab	4.2(−1.2) Bc
0.25	4.6(−0.8) Aab	4.7(−0.7) Aa	4.3(−1.1) Aab	4.2(−1.2) Bc	4.4(−1.0) Aab	4.7(−0.8) Aab
EC (mS/cm)						
0	−1.13 Aa	−0.25 Ac	−0.73 Ab	−0.43 Ac	−0.28 Ac	−0.25 Ac
0.25	−0.26 Bb	−0.21 Ab	−0.91 Aa	−0.14 Bb	−0.30 Ab	−0.26 Ab
Ca (% consumed from the media)						
0	44.5 Ac	40.2 Ad	39.5 Bd	63.9 Ab	68.9 Aa	40.5 Bd
0.25	21.4 Be	44.9 Ac	55.9 Aa	33.6 Bd	44.9 Bc	50.4 Ab
P (% consumed from the media)						
0	46.9 Ab	44.9 Ac	30.6 Be	42.9 Ad	51.0 Aa	24.5 Af
0.25	41.3 Ba	13.0 Bc	37.0 Ab	5.4 Be	10.9 Bd	13.0 Bc
Mg (% consumed from the media)						
0	17.5 Ab	14.3 Acd	8.3 Be	23.2 Aa	12.3 Bd	15.4 Abc
0.25	12.4 Bc	0.0 Bd	16.4 Ab	21.7 Aa	20.6 Aa	16.8 Ab

Different capital letters indicate a significant difference between the presence and absence of BAP in the culture medium. Lowercase letters show the differences between the light spectra, according to the Duncan test; *p* < 0.05.

**Table 4 plants-13-02842-t004:** Percentage of increase or decrease—in response to the addition of 6-BAP—of nutrient contents in in vitro tissues of shoots of *Melaleuca alternifolia* cultivated under different LED wavelengths.

LED Spectra	Nutrients in the Shoot Tissues (%)									
	P	K	Ca	Mg	S	Fe	Mn	Cu	Zn	B
White	−35.8	−25.9	+7.4	−19.9	−26.9	+21.5	−34.6	−33.8	−2.2	−29.8
Dark Conditions	+97.3	+106.9	+328.1	+181.9	+392.3	+341.6	+285.2	+62.5	+244.7	+319.5
Blue	−21.6	+29.4	−26.3	−3.3	−13.5	+20.3	−19.3	−13.0	+17.6	−24.0
Red	−33.3	+4.5	+12.4	+1.6	−17.7	+14.0	−25.1	−30.2	+12.7	+1.3
White + Red	−48.2	−12.4	−17.2	−17.0	−39.2	+66.8	−37.2	−57.4	−15.9	−28.7
Red + Blue	−36.7	+23.6	+5.1	0.0	+7.6	+3.4	−15.1	−34.5	+18.3	−39.4

A negative number means a decrease and a positive number means an increase in the percentage of the nutrient content in response to the addition of 6-BAP to the culture media, compared to the 6-BAP-free culture medium.

## Data Availability

The datasets generated during and/or analyzed during the current study are available from the corresponding author upon reasonable request.

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
