# Peer review of "Development, Chlorophyll Content, and Nutrient Accumulation in In Vitro Shoots of Melaleuca alternifolia under Light Wavelengths and 6-BAP"

_plants, 2024, doi:10.3390/plants13202842_

Round 1

Reviewer 1 Report

Comments and Suggestions for Authors

The manuscript entitled “Development, chlorophyll content and nutrient accumulation  in in vitro shoots of Melaleuca alternifolia under light-wave lengths and benzylaminopurine (BAP)” explored the effects of wavelengths on the development and physiology of melaleuca cultivated in vitro, as well as the interaction of LEDs with the BAP. However, its current presentation and style as well as English language expression should be much improved since there were demerits as listed following:
1. Line 20:“2+”in “Ca2+, Mg2+” should be superscript.

2. Line 98, 100, 205, 207, 373, 374 and 450:“-1”in “L-1” should be superscript.Check carefully throughout the manuscript.

3. Table 1: Change the unit of g/plant”into “mg/plant” and the data in the table should be changed accordingly.

4. Figure 1: Picture size for Dark conditions should be the same to the others.

5. Line 116:“2”in “mm2” should be superscript.

6. Line 118: unit is missing for “173.3”.

7. Line 189: “while the Fe concentration was higher in the presence of BAP”, why?

8. Line197: “Melaleuca alternifolia”should be italic format.

9. Line : “-1”in “L-1” should be superscript.

10. Line 207: Change “0,124”into “0.124”.

11. Line 249 and 250: unit is missing for “10.9” and “17.3”.

12. Line 253: “Cucumber seedlings” is not italic format.

13. Line 287, 288 and 291: number should be superscript.

14. Line 368: unit is missing for “1.0-1.5”.

15. Line 383-386:  “2” in “m2s” should be superscript.

16. Why 0.25 mg/L BAP was used for in vitro culture of Melaleuca alternifolia in the culture media?

17. The P, K, Ca, Mg, S, Fe, Mn, Cu, Zn and B nutrients were analyzed. How about the N macronutrient?

Comments on the Quality of English Language

Current presentation and style as well as English language expression should be much improved.

Author Response

The manuscript entitled “Development, chlorophyll content and nutrient accumulation  in in vitro shoots of Melaleuca alternifolia under light-wave lengths and benzylaminopurine (BAP)” explored the effects of wavelengths on the development and physiology of melaleuca cultivated in vitro, as well as the interaction of LEDs with the BAP. However, its current presentation and style as well as English language expression should be much improved since there were demerits as listed following:

Dear reviewer one: We would like to thanks for the contributions of you to our paper, which certainly lead to increase the quality of our manuscript to publish in Plants. We try to follow each of commentaries, responding one by one carefully. Also, we revised the English language with one English native speaker to improve the quality of English writing and style. We hope the revised paper are considered for publication in Plants. Best regards, Jean

  1. Line 20:“2+”in “Ca2+, Mg2+” should be superscript.

Response: All ions were corrected for superscript, as recommended

  1. Line 98, 100, 205, 207, 373, 374 and 450:“-1”in “L-1” should be superscript.Check carefully throughout the manuscript.

Response: All values of units were corrected as recommended

  1. Table 1: Change the unit of g/plant”into “mg/plant” and the data in the table should be changed accordingly.

Response: Changed!

  1. Figure 1: Picture size for Dark conditions should be the same to the others.

Response: Changed as recommended

  1. Line 116:“2”in “mm2” should be superscript.

Response: OK

  1. Line 118: unit is missing for “173.3”.

Response: Ok

  1. Line 189: “while the Fe concentration was higher in the presence of BAP”, why?

Response: Discussed in lines 353-363. We also improved this discussion based on studies under ex vitro conditions, since under in vitro no studies were found exploring these effects.

  1. Line197: “Melaleuca alternifolia”should be italic format.

Response: Ok

  1. Line 207: “-1”in “L-1” should be superscript.

Response: Ok

  1. Line 207: Change “0,124”into “0.124”.

Response: Ok

  1. Line 249 and 250: unit is missing for “10.9” and “17.3”.

Response: Ok

  1. Line 253: “Cucumber seedlings” is not italic format.

Response: Ok

  1. Line 287, 288 and 291: number should be superscript.

Response: Ok

  1. Line 368: unit is missing for “1.0-1.5”.

Response: Ok

  1. Line 383-386:  “2” in “m2s” should be superscript.

Response: Ok

  1. Why 0.25 mg/L BAP was used for in vitro culture of Melaleuca alternifolia in the culture media? 

Response: We appreciate your question. We have been working with Melaleuca alternifolia and learning about its behavior in the context of our research. We have determined that a concentration of 0.25 mg/L BAP effectively promotes shoot proliferation and change completely the in vitro development of melaleuca. Previously, the results from the effects of BAP was published in 2021 (https://link.springer.com/article/10.1007/s00468-021-02131-w), supporting the use of this concentration in the culture medium in contrast with no BAP addition.

  1. The P, K, Ca, Mg, S, Fe, Mn, Cu, Zn and B nutrients were analyzed. How about the N macronutrient?

Response: We appreciate your comment and recognize the importance of nitrogen (N) as a macronutrient. Unfortunately, due the small amount of tissue obtained from the treatments, the N was not quantified. The quantification of N required at least 1 g of tissue and the total amount of dry mass was 1 g, which support the evaluation of all other nutrients, except for N. In addition, we quantified the chlorophyll contents, that indirectly showed the state of N in plants. By these analysis, the concentration of chlorophyll showed a low concentration under the presence of BAP in the culture medium, in part due the lower uptake of N or the low contents of N in the tissues.

Reviewer 2 Report

Comments and Suggestions for Authors

Dear authors,

I have reviewed your manuscript with ID plants-3192537, "Development, chlorophyll content and nutrient accumulation in in vitro shoots of Melaleuca alternifolia under light-wavelengths and benzylaminopurine (BAP)". The paper is well written and I appreciate the timely subject of the paper.

This paper is important for research in the field and should be considered for publication.

Thank you for your contribution and the efforts you have put into this study. However, I would like to provide a few additional comments:

I suggest removing "benzylaminopurine" from the title and keeping only "BAP".

The work presented by Antony Cristhian Gonzales-Alvarado and Jean Carlos Cardoso is dedicated to the in vitro multiplication of Melaleuca alternifolia. Given that this plant has important pharmaceutical properties, there is significant interest in developing micropropagation protocols to assess the commercial feasibility of this technique.

This study brings novel elements to the field of micropropagation of this species by exploring the interaction between BAP and different LED wavelengths on the in vitro development of Melaleuca shoots and plantlets. Another innovative aspect addressed by the authors is the study of the main changes in culture media related to nutritional characteristics. At the same time, this research aimed to identify the best wavelengths associated with the proliferation/multiplication of Melaleuca under in vitro conditions.

The methodology is well presented, allowing the protocol to be easily reproduced in other laboratories. The tables and figures are clear and representative. The bibliography is appropriate to the study and fairly up-to-date. The conclusions are supported by the results obtained.

I would like to pose a few questions or suggestions for the authors to consider in future developments:

In fact, you studied two factors that influence the in vitro growth of Melaleuca – (1) BAP and (2) different LED wavelengths. As you have noted, the shoot length is quite small, even at the best wavelength. Do you think it would be worthwhile to test other cytokinins, given that most published articles focus on BAP?

Considering the changes observed in the basal medium, do you believe that modifying the composition of the basal medium or using other culture media (e.g., DKW) could be of interest for the micropropagation of this species?

Author Response

Dear authors,

I have reviewed your manuscript with ID plants-3192537, "Development, chlorophyll content and nutrient accumulation in in vitro shoots of Melaleuca alternifolia under light-wavelengths and benzylaminopurine (BAP)". The paper is well written and I appreciate the timely subject of the paper.

This paper is important for research in the field and should be considered for publication.

Thank you for your contribution and the efforts you have put into this study. However, I would like to provide a few additional comments:

I suggest removing "benzylaminopurine" from the title and keeping only "BAP".

Response: We change the title to accept this suggestion

The work presented by Antony Cristhian Gonzales-Alvarado and Jean Carlos Cardoso is dedicated to the in vitro multiplication of Melaleuca alternifolia. Given that this plant has important pharmaceutical properties, there is significant interest in developing micropropagation protocols to assess the commercial feasibility of this technique.

This study brings novel elements to the field of micropropagation of this species by exploring the interaction between BAP and different LED wavelengths on the in vitro development of Melaleuca shoots and plantlets. Another innovative aspect addressed by the authors is the study of the main changes in culture media related to nutritional characteristics. At the same time, this research aimed to identify the best wavelengths associated with the proliferation/multiplication of Melaleuca under in vitro conditions.

The methodology is well presented, allowing the protocol to be easily reproduced in other laboratories. The tables and figures are clear and representative. The bibliography is appropriate to the study and fairly up-to-date. The conclusions are supported by the results obtained.

Dear Reviewer 2, we would like to thanks you for commentaries and suggestions for the future experiments. The questions you send to us, are responded below with details. Also, we change the title, maintaining only the term 6-BAP, instead of complete name of this cytokinin. Thanks and best regards, Jean

I would like to pose a few questions or suggestions for the authors to consider in future developments:

In fact, you studied two factors that influence the in vitro growth of Melaleuca – (1) BAP and (2) different LED wavelengths. As you have noted, the shoot length is quite small, even at the best wavelength. Do you think it would be worthwhile to test other cytokinins, given that most published articles focus on BAP?

Response: We are very grateful for your observation and suggestion. In our study, we have focused on two crucial factors for in vitro growth of Melaleuca alternifolia: BAP and different LED wavelengths. While we have noticed that shoot length remains relatively small even under the best wavelength conditions, we believe that exploring other cytokinins, specially topolins could open up new possibilities. This idea, aligned with the fact that most studies have focused on BAP including our previous study published (https://link.springer.com/article/10.1007/s00468-021-02131-w) , would undoubtedly enrich future research. Also, we already have a history of using BAP in this species, and our previous research has demonstrated its effectiveness for in vitro multiplication, with no negative consequences to the next phases, such as rooting and elongation. This project is part of a broader effort to develop a comprehensive technology package for Melaleuca alternifolia, a species of great commercial value due to its outstanding pharmaceutical properties. We believe that the evaluation of other cytokinins could be an excellent way to further optimize it’s in vitro cultivation, mainly due the negative effects of BAP in the size and quality of shoots developed under this plant growth regulator influence.

Considering the changes observed in the basal medium, do you believe that modifying the composition of the basal medium or using other culture media (e.g., DKW) could be of interest for the micropropagation of this species?

We appreciate your suggestion on the modification of the basal medium. We believe that exploring other culture media compositions, such as DKW or WPM, could be a valuable option to optimize the micropropagation of Melaleuca alternifolia. Although in this study we focused on MS medium due our previous results obtained in (https://link.springer.com/article/10.1007/s00468-021-02131-w), the changes in nutrients during in vitro culture suggest to us that adjusting the basal composition or trying other media could further improve shoot growth and development. We find this idea interesting and it will certainly be considered in our future research, always with the goal of refining micropropagation protocols for this important species.

Reviewer 3 Report

Comments and Suggestions for Authors

The authors of the manuscript ‘Development, chlorophyll content and nutrient accumulation in in vitro shoots of Melaleuca alternifolia under light-wavelengths and benzylaminopurine (BAP)’ studied the effect of wavelengths on shoot induction response of M. alternifolia nodal explants to BAP. They also analyzed the changes in electrical conductivity, pH, and the levels of calcium, phosphorus, and magnesium in the culture medium, as well as the nutrient accumulation in the shoots. The authors focused on the quality of light but neglected to consider the quantity of light. The English text should be proofread and edited by a native speaker.

Specific comments:

Abstract:

L14: ‘The use of light-emitting diodes’ is not a recent technology.

L17-21: Please rewrite the sentences. For example, the BAP, wavelengths and their interaction had a significant effect on most of the variables analyzed.

L26: in vitro shoot tissues.

Keywords:

Please delete ‘micropropagation’ alternative keyword ‘shoot multiplication’.

Introduction:

L39: low rooting percentage [8,9]. Please replace the reference 9 with the following:

Guo Y (2007) Technology of cuttage seeding-raising of Melaleuca alternifolia. Prot For Sci Technol. http://en.cnki.com.cn/Article_en/CJFDTotal-FHLK200703010.htm

Uchoi A, Kumar N, Rajamani K, Sumitha S (2018) Effect of plant growth regulators on vegetative and seed propagation of tea tree (Melaleuca alternifolia L.). Int J Chem Stud 6(2):468–472.

L54-55: Please provide more information on the effect of different wavelengths on in vitro shoot production such as shoot formation percentage, number of shoots, and shoot biomass.

L60: Please indicate the type of LEDs used. Reference 9: authors used white and red (1:1) light-emitting diode (LED).

L66-67: Citation required.

L73: plantlets? BAP induces the growth of multiple shoots in this species but does not promote rooting.

Results:

L79: Please rewrite.

L80: Please indicate the level of significance.

L90: ‘BR’ light. Please expand the abbreviation at the first use (throughout the text).

L92-93: ‘the highest fresh and dry masses of plants (0.18 g/plant and 0.015 g/plant,

respectively) were obtained using white light in the presence of BAP’ Plant? Did the authors mean explant? Please use appropriate terms.

L98-100: Correct the units (typo) throughout the text.

L100: light spectra instead of wavelengths.

L114: Did the authors observe any changes in the morphology of stomata? Please provide supporting figures for better understanding.

L176: Figure 2. Correct pH.

L196: Please indicate the nutrient contents in shoot tissue in µg/g dry weight for better understanding. Because the authors used ICP to determine the nutrient contents.

Discussion:

L213-215: BAP is used to induce multiple shoots, not rooting. The micro shoots regenerated were transferred to shoot elongation medium and then subjected to rooting (L219-221). The harmful effects of BAP are abnormal shoots, vitrification etc.

Materials and Methods:

L371: modified Murashige and Skoog (MS) medium

L378: Please indicate the number of nodes in each segment. Also, the size of the explant.

L381: Please provide the spectrum of all wavelengths for better understanding.

L383-386: The light intensity for each treatment differs, which also affects shoot development.

L426-428: Please provide details on how the authors measured pH and EC of the semi-solid media.

L432-437: Please provide detailed methods for analyzing macronutrients in the culture media as a supplementary file.

Comments on the Quality of English Language

Extensive editing of English language required.

Author Response

Reviewer 3

The authors of the manuscript ‘Development, chlorophyll content and nutrient accumulation in in vitro shoots of Melaleuca alternifolia under light-wavelengths and benzylaminopurine (BAP)’ studied the effect of wavelengths on shoot induction response of M. alternifolia nodal explants to BAP. They also analyzed the changes in electrical conductivity, pH, and the levels of calcium, phosphorus, and magnesium in the culture medium, as well as the nutrient accumulation in the shoots. The authors focused on the quality of light but neglected to consider the quantity of light. The English text should be proofread and edited by a native speaker.

Dear Reviewer 3, We would like to thanks for your efforts and strong suggestions aimed to improve our manuscript. We try to respond and agree with your valuable suggestions and corrections suggested. We also send the reviewed paper to a native english speaker for improve english language. Thanks and best regards, Jean 

Specific comments:

Abstract:

L14: ‘The use of light-emitting diodes’ is not a recent technology.

Response: Ok, we changed this affirmative in the text. Our interest is to show that although led light is used in commercial labs, this is recent and the wavelength and intensity of this lights is not completelly understood, especially in terms of efficiency for cultivation and their relation with other factors of in vitro cultivation, such their interaction with plant growth regulators. However, we follow your suggestion to make this sentence more precise.

L17-21: Please rewrite the sentences. For example, the BAP, wavelengths and their interaction had a significant effect on most of the variables analyzed.

Response: Thanks for your suggestion and we did this change in the text.

L26: in vitro shoot tissues.

Response: Ok

Keywords:

Please delete ‘micropropagation’ alternative keyword ‘shoot multiplication’.

Response: Changed

Introduction:

L39: low rooting percentage [8,9]. Please replace the reference 9 with the following:

Guo Y (2007) Technology of cuttage seeding-raising of Melaleuca alternifolia. Prot For Sci Technol. http://en.cnki.com.cn/Article_en/CJFDTotal-FHLK200703010.htm

Uchoi A, Kumar N, Rajamani K, Sumitha S (2018) Effect of plant growth regulators on vegetative and seed propagation of tea tree (Melaleuca alternifolia L.). Int J Chem Stud 6(2):468–472.

Response: These references was replaced and included in our manuscript.

L54-55: Please provide more information on the effect of different wavelengths on in vitro shoot production such as shoot formation percentage, number of shoots, and shoot biomass.

Response: We add the following paragraph to the text:

Currently, the effects of LEDs with different wavelengths (colors) are largely explored on the in vitro development of diverse plant species [18]. In Vitis riparia, a significant increase in shoot height was observed when using red light, which also resulted in an increase in shoot biomass [19]. In Helianthus tuberosus, blue light favored the increase in biomass and overall the growth of explants [20], while in Gossypium hirsutum, Cedrela fissilis and Pinus pseudostrobus the combination of red and blue light increased shoot production, elongation and biomass accumulation [21, 22, 23]. Since these different results reported, the establishment of optimal conditions for in vitro cultivation using LEDs are in growth. Also, the most of studies with in vitro micropropagation of Melaleuca have used conventional cold lamps or LEDs with specific wavelengths of white and red/blue light combination (1:1), respectively [11,12,24].

L60: Please indicate the type of LEDs used. Reference 9: authors used white and red (1:1) light-emitting diode (LED).

Response: Ok, the references were added.

L66-67: Citation required.

Response: reference added

L73: plantlets? BAP induces the growth of multiple shoots in this species but does not promote rooting.

Response: Ok, replaced

L79: Please rewrite.

Response: The sentence was rewritten

L80: Please indicate the level of significance.

Response: Ok

L90: ‘BR’ light. Please expand the abbreviation at the first use (throughout the text).

Response: Ok

L92-93: ‘the highest fresh and dry masses of plants (0.18 g/plant and 0.015 g/plant, respectively) were obtained using white light in the presence of BAP’ Plant? Did the authors mean explant? Please use appropriate terms.

Response: The term explant was replaced by shoot.

L98-100: Correct the units (typo) throughout the text.

Response: Ok, corrected

L100: light spectra instead of wavelengths.

Response: Ok, this term was replaced along the text

L114: Did the authors observe any changes in the morphology of stomata? Please provide supporting figures for better understanding.

Response: We appreciate your comment. Unfortunately, we are unable to include a figure of the stomata, as the available images do not have adequate resolution to illustrate the morphological changes clearly, which difficulty also the visual interpretation of other stomata morphological changes, we prefer do not add to this manuscript.

L176: Figure 2. Correct pH.

Response: Ok

L196: Please indicate the nutrient contents in shoot tissue in µg/g dry weight for better understanding. Because the authors used ICP to determine the nutrient contents.

Response: Dear reviewer, thanks for your commentary. We added a table (Table S1) containing the information about nutrient contents in the shoots, as requested. Yes, the analysis was realized using ICP.

Discussion:

L213-215: BAP is used to induce multiple shoots, not rooting. The micro shoots regenerated were transferred to shoot elongation medium and then subjected to rooting (L219-221). The harmful effects of BAP are abnormal shoots, vitrification etc.

Response: Thanks for this suggestion. We rewritten this sentence for better meaning, as follow

However, BAP had harmful effects on in vitro melaleuca development, strong decreasing shoot length and height and root inhibition in shoots (Figure 1). Despite these negative effects also reported by Iiyama and Cardoso [11], these authors observed that BAP-derived microshoots developed normally, when subcultured in the rooting/elongation medium in the absence of BAP.

Materials and Methods:

L371: modified Murashige and Skoog (MS) medium

Response: Ok

L378: Please indicate the number of nodes in each segment. Also, the size of the explant.

Response: This information was added: The test tubes containing individual microcuttings, each one with four nodes and 0.7±0.1 cm (replicates),

L381: Please provide the spectrum of all wavelengths for better understanding.

Response: We was added the most of spectra used, except for one we not have the spectra

(Ourolux®, São Paulo, Brazil), with peaks at 440–450 nm (blue), 540–550 nm (green) and 610–620 nm (red), W: 64.09 μmol/m-2 s-1), 2) dark conditions (D: 0.58 μmol/m2s – diffuse light), 3) 100% blue LED light (Phillips Greenpower LED Research Module Blue, ~440 nm, Pila, Poland, B: 55.58 μmol/m2s), 4) 100% red LED light (Phillips Greenpower LED module HF deep red, ~660 nm, Pila, Poland, R: 106.22 μmol/m2s), 5) 50%: 50% red and white LED light (RW: 104.51 μmol/m2s), and 6) 40%: 60% blue and red LED light (LabPar, with wavelength peaks at 447 nm, range 420–470 nm—blue and 667 nm, range 625–680 nm—red, Barueri, Brazil, BR: 71.48 μmol/m2s).

L383-386: The light intensity for each treatment differs, which also affects shoot development.

Response: Dear reviewer, thanks for your comment and we agree that light intensity (specially in PPFD) also have important influences in shoot development. We analyzed this possible difference due the differences in PPFD reported according different light spectra, but we not observed strong correlations between the PPFD and the most of observations we realized in the in vitro shoot development. As example, the better responses for each variable depends much more of the light spectra than the PPFD: the blue leds (55,58 umol) have the low PPFD values, followed by white LEDs (64,1 umol), but we had the best responses using white and worst in blue. The use of highest PPFDs by Red and R+W light spectra resulted in intermediary values for shoot development. Thus, although there are a large range of PPFD according the light spectra, there are no correlation between the increase of PPFD and the responses of shoot development.

L426-428: Please provide details on how the authors measured pH and EC of the semi-solid media.

Response: We appreciate the valuable suggestion. The follow information was added: For these analyses, the culture media are previously frozen at -20°C for 24 hours, followed by defrost and separation of liquid phases for the analysis using the pHmeter and conductivity meter.

L432-437: Please provide detailed methods for analyzing macronutrients in the culture media as a supplementary file.

Response: We welcome your comments. We added the information about the manual of instructions provided by Hanna Instruments, as follow:

For the analysis of the consumption of the macronutrients: Ca2+ - using 3 mL of sample, 10 mL of reagent HI93752A-Ca, 4 drops of buffer and 1 mL of HI93752B-Ca, reading on the absorbance at 466 nm; Fosfate (High range) – 10 mL of sample, ten drops of HI93717A-0 and one packet of HI93717B-0, reading the absorbance at 525 nm, and; Mg2+ - using 1 mL of HI93752A-Mg, 10 mL of HI93752B-Mg and 0.5 mL of sample, reading the absorbance at 466 nm. The sample used is from the culture media, the initial and final values of these nutrients were also analysed, aiming to determine how much of these nutrients are uptake from the culture media. The concentrations (mg L-1) of these nutrients from the culture media were obtained using the equipment IRIS-HI801 Spectrophotometer (Hanna Instruments, Portugal), using the methodologies and solutions provided by the manufacturer (HI801 Manual of instructions, Hanna Instruments, 2020, Rhode Island, USA)

Round 2

Reviewer 3 Report

Comments and Suggestions for Authors

"We added a table (Table S1) containing the information about nutrient contents in the shoots, as requested. The supplementary file is missing. Please remember to upload the supplementary file and include the details in the manuscript.

Comments on the Quality of English Language

Minor editing of English language required.